# Behavior of Muscle-Derived Stem Cells on Silica Nanostructured Substrates

**DOI:** 10.3390/nano10091651

**Published:** 2020-08-22

**Authors:** Hyo-Sop Kim, Bit Na Lee, Sangdun Choi, Moon Suk Kim, Jae-Ho Kim

**Affiliations:** Department of Molecular Science and Technology, Ajou University, Suwon 16499, Korea; ikari06@ajou.ac.kr (H.-S.K.); newri21@naver.com (B.N.L.); sangdunchoi@ajou.ac.kr (S.C.)

**Keywords:** silica nanostructured substrates, rat muscle-derived stem cells, nanoscale, adhesion, proliferation

## Abstract

The aim of the present work was to evaluate the responses of rat muscle-derived stem cells (rMDSCs) to growth on silica nanostructured substrates (SN) with nanoscale topographic surfaces. SN of different sizes (SN-60, SN-150, SN-300, SN-500, and SN-700) were prepared using silica nanoparticles with sizes of 60–700 nm. The prepared SN showed roughness at the nanoscale level. The total number of adherent cells on SN increased with increasing nanoscale level and incubation time. The rMDSCs attached to SN-500 and SN-700 were extensively flattened, whereas those grown on SN-60, SN-150, and SN-300 were more rounded. The rank order of the cell length and height of attached rMDSCs at 5 d on different surfaces was SN-60 ≈ SN-150 >> SN-300 > SN-500 > SN-700 > glass. Compared with rMDSCs grown on SN-60, SN-150, or SN-300, those attached to SN-500 and SN-700 exhibited a distinct morphology with filopodial extensions and stronger expression of focal adhesion, integrin, and actin. An evaluation of the gene expression of adhered rMDSCs showed that rMDSCs grown on SN-300 exhibited a higher environmental stress response than those grown on glass or SN-700. Collectively, our data provide fundamental insight into the cellular response and gene expression of rMDSCs grown on nanostructured substrates.

## 1. Introduction

Several studies have tested cellular responses to nanoscale topographic substrates, showing that the contact of different human tissues with nanostructured substrates can influence cellular responses in biomedical applications [1,2,3,4]. Studies performed to date have reported that various nanostructured substrates can modulate the adhesion and proliferation of different mammalian cell types [5,6]. Recent studies have shown that topographic features of nanostructured substrates, such as their size and spatial arrangement, play a key role in basic cellular responses, such as adhesion and proliferation. The acknowledgment of the differential responses of cells to various nanostructured substrates has motivated intense research efforts into understanding the basic mechanisms involved in cellular responses to nanostructured substrates [7,8].

Several methods for fabricating various nanostructured substrates have been developed, including immersion, vapor deposition, wet etching, reactive ion etching, scanning tunneling microscopy, photoimmobilization, microfluidic techniques, and Langmuir–Blodgett (LB) techniques [9]. Of these methods, LB techniques are the easiest way to deposit single layers on a solid substrate [10,11,12].

Silica substrates are currently being widely used in biomedical applications [13]. However, because the silica substrates reported to date have relatively limited biocompatibility, there is a lack of understanding of the cellular responses of mammalian cells to growth on silica substrates [14,15]. In previous studies, our group fabricated silica nanostructured substrates (SN) on glass using the LB technique, demonstrating that this approach provides a simple method for producing uniform and reproducible SN with nanoscale level roughness [16,17,18]. We also examined SN as culture substrates for chondrocytes and observed that SN promoted good adhesion and proliferation without promoting dedifferentiation and thus served as adaptable culture substrates for chondrocytes [18]. Although chondrocytes were found to adhere, spread, and proliferate on SN, additional detailed studies of cellular responses of other mammalian cells grown on SN are needed.

Stem cells, which are self-renewing and have the inherent capacity to differentiate into many cell types, have emerged as an important resource for biomedical fields, such as cell therapy [19,20]. Adult stem cells can be harvested from various sources, including bone marrow, adipose tissue, human umbilical cord, and muscles. Of these, skeletal muscle-derived stem cells (MDSCs) have recently attracted attention as a potential source of adult stem cells owing to the considerable mass of skeletal muscle in the body [21,22]. In addition, MDSCs are central to the maintenance of body tissues. Importantly, however, the mechanism by which MDSCs respond to SN fabricated from silica substrates that are widely used in biomedical applications is unknown.

In general, cells attach to their substrates via focal adhesion with integrins linked to the cell cytoskeleton [23]. Actin, which is linked to α-actinin and to the membrane through vinculin, can participate in cellular processes, such as establishment and maintenance of cell shape [24]. Integrin β1, which is a well-known transmembrane glycoprotein, can form a functional interaction network with the substrate and can mediate several functions, such as cell attachment and proliferation on a substrate [25]. Heat shock protein 70 (HSP70), which is a well-known anti-apoptotic protein, is triggered in response to environmental stress [26].

In the current study, we investigated the adhesion and proliferation of rat MDSCs (rMDSCs) grown on SN with roughness controlled at the nanoscale level. Another objective of this work was to investigate the associated changes in proteins known to be involved in adhesion by immunostaining for actin, focal adhesions, and integrin β1 and by examining β-actin and integrin α8 mRNA expression using reverse transcription-polymerase chain reaction (RT-PCR). In addition, stress responses associated with the adhesion and proliferation on SN were evaluated by examining changes in HSP70 expression.

## 2. Materials and Methods

### 2.1. Materials

Tetraethylorthosilicate (TEOS, TCI, Tokyo, Japan), absolute ethanol (HPLC, Fisher scientific, Waltham, MA, USA), deionized water (18.2 M, Milli Q water purification system, Merck, Darmstadt, Germany), and 25% aqueous solution of ammonium hydroxide (JUNSEI, Tokyo, Japan) were used for the synthesis of silica nanoparticles.

### 2.2. Preparation of SN

Monodispersed silica nanoparticles (60, 150, 300, 500, and 700 nm) were prepared using the Stober method [27]. The silica nanoparticles were purified using a centrifugation and redispersion procedure in ethanol. These nanoparticles were then treated with 4-amino benzene thiol and dispersed in chloroform. Each silica nanoparticle was transferred onto a common microcover glass (12 mm, circle, Matsunami, Osaka, Japan) to perform a Langmuir trough (Nima Technologies, Manchester, UK). The silica substrate fabricated with an area of 100 cm^2^ was heated in an electric furnace at 550 °C. After 5 h, the silica substrate was cleaned for 2 h using a UV/ozone cleaner (Jelight, Irvine, CA, USA). The SN prepared by silica nanoparticles of 60, 150, 300, 500, and 700 nm were designated as SN-60, SN-150, SN-300, SN-500, and SN-700, respectively. The prepared SN were characterized by atomic force microscopy (AFM) measurements in non-contact mode with an acquisition frequency of 312 kHz using an atomic force microscope (XE-100, Park systems, Suwon, Korea).

### 2.3. Rat Muscle-Derived Stem Cell Isolation

Primary muscle-derived cells were purified as described previously [21,22]. The rat’s main hind-limb muscles were removed; a lot of connective tissue and fat were cut out, and the obtained muscles were trimmed by hand and then washed twice with potassium bisulfate (PBS, pH 7.4). The obtained pellets were centrifuged at 2000 rpm for 5 min and enzymatically dissociated by adding 0.2% collagenase-type XI (Sigma, St. Louis, MO, USA) for 1 h at 37 °C, dispase (20 mL, Sigma, Tokyo, Japan) for 45 min, and 2× trypsin for 30 min. Cells were separated from muscle fiber fragments and tissue debris by differential centrifugation and then plated on collagen-coated flasks in Dulbecco’s modified Eagle’s medium (DMEM) containing 10% fetal bovine serum (HyClone, Tauranga, New Zealand), 100 U/mL penicillin, and 100 μg/mL streptomycin for 1 h. The muscle cell extract was pre-plated on collagen-coated flasks. After 1-h incubation, non-adherent cells suspended in the medium were transferred to fresh collagen-coated flasks. The subsequent pre-platings were performed in the next 2 h, 3 h, 1 day, 2 days, and 3 days. Finally, the rMDSCs were seeded into normal tissue culture flasks at 1 × 10^5^ cells/cm^2^. The flasks were rinsed three times with PBS on the second day of expansion. The medium was changed every 2 days throughout the study period. Adherent cells, rMDSCs, were rinsed thoroughly with PBS and detached using 0.05% trypsin-ethylenediaminetetraacetic acid for experimental use. The final rMDSCs used were taken at passage 5.

### 2.4. Culture of rMDSCs on SN

The SN were dipped in 99% ethanol for 24 h and placed in a 24-well plate (SPL, Gyeonggi, Korea). The rMDSCs (2 × 10^4^ cells/well) were plated in the well plate with SN and incubated in culture medium (Dulbecco’s modified Eagle’s medium (DMEM) supplemented with 5% fetal bovine serum, 5% horse serum, and 2% penicillin) at 37 °C in a humidified incubator containing 5% CO_2_. The culture medium was replaced every 2 days. To detect the morphology of the rMDSCs on SN, digital images in the culture medium were captured using an inverted phase contrast microscope (ECLIPSE TS100, Nikon, Tokyo, Japan) with iWorks software (Nahwoo, Gyeonggi, Korea) and a confocal laser microscope (OLS 3000, Olympus, Tokyo, Japan) under atmospheric conditions. rMDSC viability on SN in the three-well plates was individually determined by MTT (3-(4,5-dimethylthiazol-2-yl)-2,5-diphenyl tetrazolium bromide) assay and then calculated as an average value. The experimental procedure was as follows: 100 μL of the MTT substrate (5 mg/mL in PBS) was added to rMDSCs cultured on the SN of each well, and the plates were incubated for 4 h at 37 °C; the resulting formazan precipitate was solubilized by the addition of 500 μL of DMSO and then shaken for 30 min. The solutions (100 μL) from each well were transferred to 96-well plates. The optical density of the transferred solutions was determined at 450 nm using a microplate reader (EL808 ultra microplate reader, Bio-Tek Instrument, Winooski River, VT, USA). All experiments were performed 3 times, and the results were presented as mean ± standard deviation. The relative percentage of cells was converted from the optical density of each SN based on the optical density of glass (control).

### 2.5. Immunocytochemistry Analysis

rMDSCs on SN were fixed with 4% para-formaldehyde (Biosesang, Gyeonggi, Korea) and washed with PBS and 1% Triton-X 100. The fixed cells were blocked using 1% bovine serum albumin (Bovogen, Essendon, Australia) at room temperature for 30 min. The fixed cells were incubated with a 1:500 dilution of anti-vinculin antibody (Sigma, St. Louis, MO, USA) for actin–FA and integrin β1 (1:200; Abcam, Cambridge, UK) for actin–integrin β1 in 1% BSA blocking solution for 16 h at 4 °C and then washed again with PBS. The fixed cells were incubated with a 1:500 dilution of anti-vinculin antibody (Sigma, St. Louis, MO, USA) for actin–FA and integrin β1 (1:200; Abcam, Cambridge, UK) for actin–integrin β1 in 1% BSA blocking solution for 16 h at 4 °C and then washed again with PBS. The fixed cells were subsequently incubated at room temperature in the dark with a solution of anti-rabbit immunoglobulin G antibody (1:1000 dilution; Abcam, Cambridge, MA, USA), washed with PBS-T (137 mM NaCl, 1.8 mM KH_2_PO_4_, 10 mM Na_2_HPO_4_, 27 mM KCl, 0.1% Tween 20, pH 7.4), and then incubated with the secondary antibody (Alexa Fluor 594 phalloidin; Life Technologies, CA, USA) for 3 h at room temperature in the dark. After 3 h, the cells were washed with PBS and then counterstained with diamino-2-phenylindoadihydrochloride (DAPI; Sigma, St. Louis, MO, USA) and washed again with PBS, and then, they were mounted with a fluorescent mounting solution (DAKO; Carpinteria, CA, USA). Immunofluorescent images were obtained using an Axio Imager A1 (Carl Zeiss Microimaging GmbH, Göttingen, Germany) and analyzed using Axiovision Rel. 4.8 software (Carl Zeiss Microimaging GmbH, Göttingen, Germany).

### 2.6. Gene Expression

RNA was isolated from rMDSCs on SN (300 and 700 nm) at 5 days with the TRIzol Reagent (Invitrogen, Carlsbad CA, USA). RNA was quantified by Nanodrop analysis (ND-100 UV/Vis Spectrophotometer; Thermo scientific, Waltham, MA, USA) at 260 nm. The extracted RNA samples (50 ng) were then reverse-transcribed using the SuperScript™ II RT (Invitrogen, Carlsbad, CA, USA). The PCR was performed using SYBR Green (N′,N′-dimethyl-N-[4-[(E)-(3-methyl-1,3-benzothiazol-2-ylidene)methyl]-1-phenylquinolin-1-ium-2-yl]-N-propylpropane-1,3-diamine) (Applied Biosystems, Warrington, UK). The amplification was carried out for 40 cycles (reaction at 94 °C for 2 min, denaturation at 94 °C for 20 s, and annealing and extension at 65 °C for 1 min).

The following primer pairs (Bioneer, Daejeon, Korea) were obtained from the published sequences: for glyceraldehyde-3-phosphate dehydrogenase (GAPDH) (153 bp), 5′-GCAAGTTCAACGGCACAGTCAAG-3′ and 5′-ACATACTCAGCACCAGCATCACC-3′; for β-actin (74 bp), 5′-GGAAATCGTGCGTGACATTAAA-3′ and 5′-GCGGCAGTGGCCATCTC-3′; for integrin α8 (109 bp), 5′-GGGACAGTAGTAGACAGC-3′ and 5′-GGACTTCTACATACCTGAT-3′; and for HSP70 (80 bp), 5′-GGTTGCATGTTCTTTGCGTTTA-3′ and 5′-GGTGGCAGTGCTGAGGTGTT-3′.

Data analysis was performed by the comparative Ct method (2-DDCt) using Opticon 2 software (Biorad, Hercules, CA, USA) for relative quantification. All samples were analyzed in triplicate. Their expression values were normalized by GAPDH expression of rMDSCs on SN.

### 2.7. Statistical Analysis

Cytotoxicity, the length and height of rMDSCs, and gene expression data were obtained from independent experiments in three wells, with data given as the mean and standard deviation (SD). The results were analyzed by one-way ANOVA with Bonferroni’s multiple comparison.

## 3. Results and Discussion

### 3.1. Preparation of SN

First, silica nanoparticles with uniform diameters of 60, 150, 300, 500, and 700 nm (± 7 standard deviations) were prepared using the Stober method (Figure 1a) [27], which creates SN on a glass surface from silica nanoparticles using the LB technique. Large areas (~100 cm^2^) with uniform roughness, which could be controlled by choosing the appropriate silica nanoparticle, were easily prepared using this approach. The resulting SN formed completely and densely packed silica surfaces with root-mean-squared roughness values of 30–490 nm, compared to a value of 5 nm for a glass substrate (Figure 1a). The prepared SN were thus well-defined characteristic topographic substrates with uniform roughness at a nanoscale level. Figure 1b shows a representative AFM image of SN-500 (AFM images of SN-60, SN-150, SN-300, and SN-700 are shown in Appendix A).

### 3.2. Adhesion and Proliferation of rMDSCs on SN

To evaluate the ability of SN to support cell attachment and proliferation, rMDSCs were cultured on all SN, monitored for optical density using MTT assays, and calculated as the relative cell numbers (Figure 2). For comparison, identical experiments were carried out on glass plates. The viability of rMDSCs grown on all SN was evaluated at different time points over a period of 9 days. The number of rMDSCs generally increased on both SN and glass plates. After 1 days, there were no significant differences in the relative numbers of adherent cells among different SN, but the relative number of adherent cells on SN was approximately 60–70% of that on glass substrates. However, at 5 days, the relative number of adherent cells on SN-60 was ~50% of that on SN-700, whereas the relative number of adherent cells on SN-700 was ~60% of that on glass. The relative number of adherent cells on SN increased with increasing size of silica nanoparticles and increasing culture time. This was in agreement with the notion that the larger the size of silica nanoparticles, the better the cell adhesion and proliferation, which is the result of Oh et al. [28].

Optical microscopic images of rMDSCs grown on SN (Figure 3) showed that after culturing for 1 day, rMDSCs were difficult to observe on SN-60, SN-150, and SN-300. In most cases, attached rMDSCs were detected on these surfaces only after 5 days and exhibited a round shape. Few filopodia were evident even after 9 days, and they did not form any branches or planar enlargements. In contrast, rMDSCs grown on SN-500, SN-700, and glass plates formed many filopodia and adhered to the substrate even after 1 d (Figure 3d–f). rMDSCs showed some flat, spread-out areas that increased contact with the substrates, and most filopodia on these rMDSCs were highly branched.

Figure 4 shows confocal images of rMDSCs grown on SN. The red and blue in confocal microscopic images correspond to higher and lower vertical profiles (cell heights), respectively, of the substrate-attached rMDSCs. At 1 day, rMDSCs had absolutely no contact with the surface of SN-60, SN-150, or SN-300. However, rMDSCs became attached to SN-500 and SN-700 even after 1 day. The majority of the attached rMDSCs on SN-500 and SN-700 at 1 day exhibited a red image, indicating a higher vertical profile, but adopted a more flattened morphology at 5 days, as evidenced by a bluer coloration in confocal images. Generally, images became bluer with increasing incubation time and nanoscale size of SN. The graph in Figure 4 indicates the average height of rMDSCs attached on all SN. rMDSCs grown on SN-500 and SN-700 exhibited lower heights even at 1 day, and their heights decreased with increasing incubation time as they became attached over a larger area. The glass plate showed a pattern similar to that of SN-500 and SN-700. In contrast, rMDSCs grown on SN-60 and SN-150 did not attach at 1 day or 2 days and showed higher cellular profiles at 5 days, indicating a more rounded morphology. The rank order of attached rMDSCs’ height at 5 days was SN-60 ≈ SN-150 >> SN-300 > SN-500 > SN-700 > glass. The rank order of attached rMDSCs’ length showed an opposite trend to the order of height.

Collectively, these results indicate that controlling SN roughness at the nanoscale level can produce an optimal roughness (i.e., SN-500 and SN-700) that promotes widely spread and flattened rMDSCs, implying that rMDSCs can recognize the nanoscale roughness to form larger adhesion sites.

### 3.3. Adhesion and Stress-Related Protein/Gene Expression in rMDSCs Grown on SN

To further examine the influence of surface topography on cellular responses of rMDSCs, we examined rMDSCs grown on SN by immunostaining for actin, focal adhesion, and integrin β1. Figure 5 and Figure 6 show pseudocolor images of actin–focal adhesion and actin–integrin β1 immunostaining obtained from rMDSCs cultured on different SN, clearly showing immunofluorescence coloration assignable to the nucleus (blue), actin (red), focal adhesion (FC, green), and integrin (green). Immunostained images were consistent with the results of optical and confocal images described in the previous section, showing large areas of vinculin and integrin staining in rMDSCs grown on SN-500 and SN-700. At 9 days, actin cytoskeleton staining revealed that rMDSCs on SN-500 and SN-700 exhibited a spindle-like shape with many filopodial extensions of different lengths. With increasing culture time, the actin cytoskeleton of rMDSCs on SN became more elongated and mature, revealing a relatively high proportion of organized bundles of actin fibers. Vinculin, integrin, and organized actin fibers were also observed in rMDSCs grown on glass.

In contrast, immunostained images showed few rMDSCs attached on SN-60, SN-150, and SN-300 at 1 day, and even at 5 days or 9 days. The attached rMDSCs that were detected exhibited a stellate shape with a few short filopodial extensions and lower intensity staining for the focal adhesion proteins vinculin (green) and actin (red). The morphology of these cells was similar to that of rMDSCs attached on SN-500, SN-700, and glass observed as early as 1 day. These results suggest that favorable substrates can be recognized by rMDSCs via actin-, focal adhesion-, and integrin-mediated binding.

To further illustrate differences in cellular responses of rMDSCs grown on SN, we analyzed the expression of actin, integrin, and HSP70 by quantitative RT-PCR using SN-300 and SN-700 as representative samples and glass as a control. As shown in Figure 7, expression of the actin gene at 5 days was only slightly higher in rMDSCs grown on SN-700 and glass, indicating no large SN-related gene difference with respect to the establishment and maintenance of cell shape. At the same time, the expression levels of integrin were significantly increased in rMDSCs grown on SN-700 and glass, indicating that rMDSCs can form a functional interaction network with SN-700 and glass, resulting in greater attachment to and proliferation on these substrates. Finally, HSP70 expression was significantly higher in rMDSCs grown on SN-300 than in rMDSCs grown on SN-700 or glass, indicating that SN-300 triggers a greater environmental stress response in rMDSCs than does glass or SN-700. This result suggests that as the roughness of the substrate is reduced, the attachment and proliferation of the rMDSCs becomes limited, resulting in greater stress on rMDSCs to attach to the surface of SN-300.

## 4. Conclusions

In conclusion, SN with nanoscale roughness affect the cellular responses of rMDSCs. An increase in nanoscale roughness was associated with increased attachment and proliferation of rMDSCs. We also found that SN affected the cytoskeletal structure, promoting an elongated cell morphology in rMDSCs. rMDSCs expressed higher levels of the stress-associated gene HSP70 on SN-300 than on glass or SN-700. Because nanoscale-controlled SN can induce adequate attachment and proliferation of rMDSCs, we believe that the SN fabricated in this study will be useful for ascertaining the relationships between substrates and cellular responses and gene expression. Further investigations of the cellular responses of other stem cells to SN are currently ongoing in our laboratory. Moreover, cellular responses of other stem cells to combinations of various SN will be assessed in future studies.

## Figures and Tables

**Figure 1 nanomaterials-10-01651-f001:**
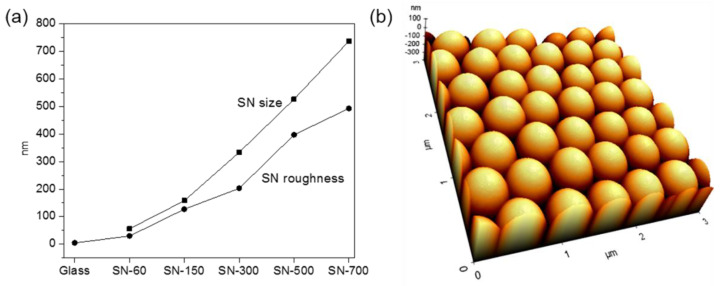
(**a**) Sizes of silica nanoparticles and roughness of silica nanostructured substrates (SN); (**b**) atomic force microscopy (AFM) image of SN-500.

**Figure 2 nanomaterials-10-01651-f002:**
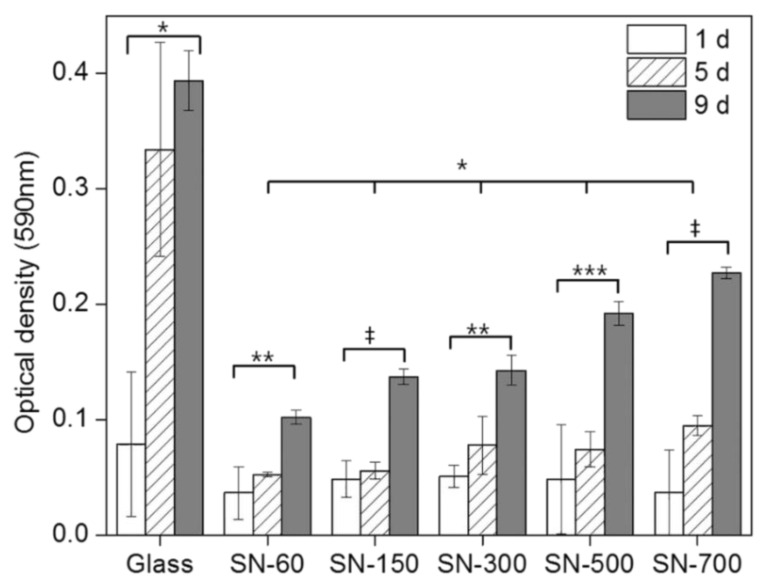
Viability of rat muscle-derived stem cells (rMDSCs) grown on SN and glass at 1, 5, and 9 days, measured by MTT assay. Cytotoxicity of substrates was analyzed by one-way ANOVA with Bonferroni’s multiple comparison (* *p* < 0.05, ** *p* < 0.01, *** *p* < 0.005, ^‡^
*p* < 0.001).

**Figure 3 nanomaterials-10-01651-f003:**
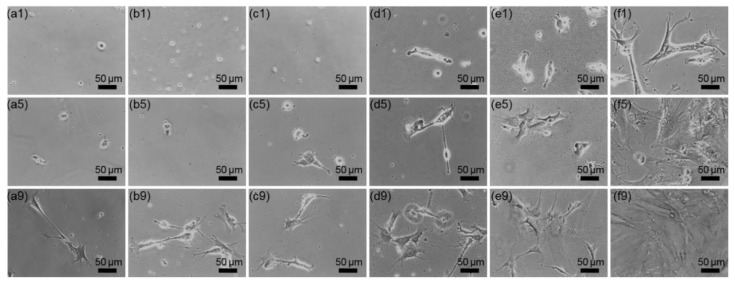
Morphology of rMDSCs on SN at 1, 5, and 9 d. (**a**) SN-60, (**b**) SN-150, (**c**) SN-300, (**d**) SN-500, (**e**) SN-700, and (**f**) glass. The numbers indicate 1, 5, and 9 d. Magnification, ×200; scale bar = 50 μm.

**Figure 4 nanomaterials-10-01651-f004:**
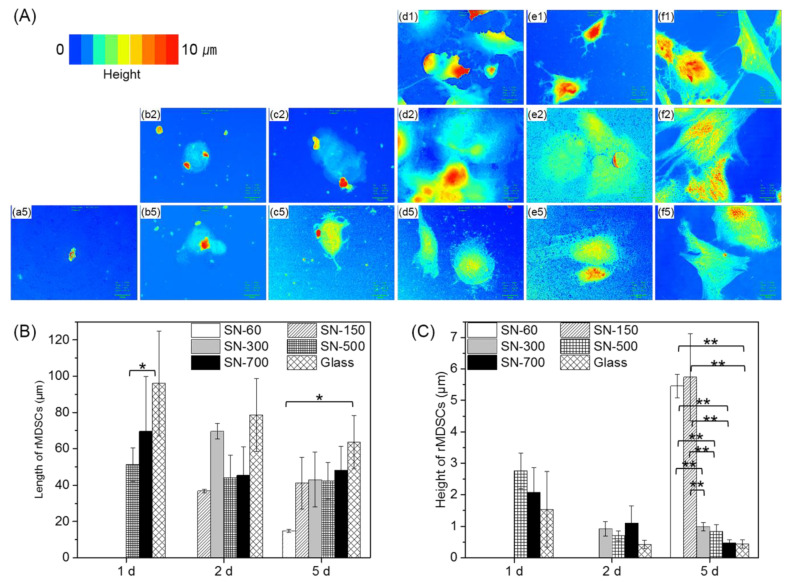
Confocal microscopic analysis of the length and height of rMDSCs grown on different SN and glass at 1, 2, and 5 days. (**A**) Confocal microscopic images; color corresponds to cell height from low (blue) to high (red). (**a**) SN-60, (**b**) SN-150, (**c**) SN-300, (**d**) SN-500, (**e**) SN-700, and (**f**) glass. The numbers indicate 1, 2, and 5 days. Magnification, ×1000; scale bar = 15 μm. Quantification of the results of (**B**) length and (**C**) height of rMDSCs shown in (**A**). Data of length and height of rMDSCs were analyzed by one-way ANOVA with Bonferroni’s multiple comparison (* *p* < 0.01, ** *p* < 0.001).

**Figure 5 nanomaterials-10-01651-f005:**
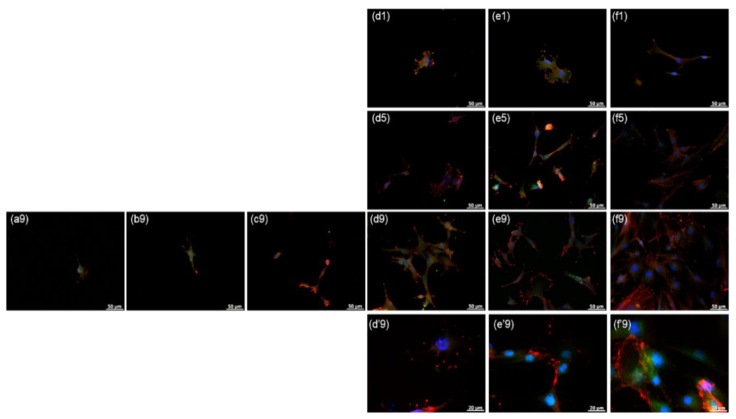
Actin–focal adhesion immunostaining in rMDSCs grown on SN for 1, 5, and 9 days. (**a**) SN-60, (**b**) SN-150, (**c**) SN-300, (**d**) SN-500, (**e**) SN-700, and (**f**) glass. The numbers indicate 1, 5, and 9 days. Diamino-2-phenylindoadihydrochloride (DAPI) (blue), actin (red); focal adhesion (vinculin; green). Magnification, ×400; scale bar = 50 μm. (**d’9**), (**e’9**), and (**f’9**) are enlarged images (magnification, ×1000). (The enlarged actin–focal adhesion immunostaining images are shown in Appendix A).

**Figure 6 nanomaterials-10-01651-f006:**
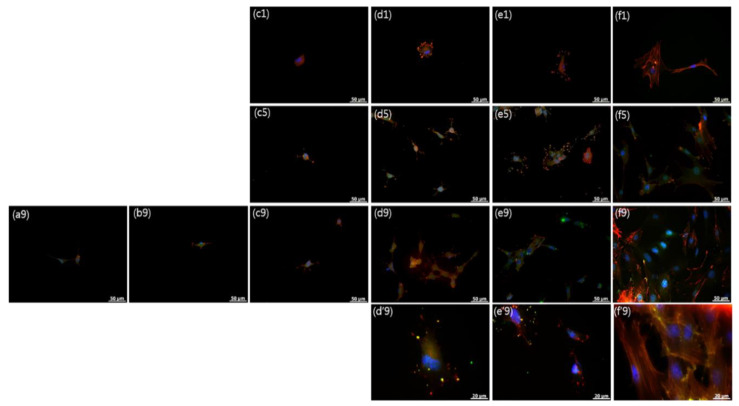
Actin–integrin β1 immunostaining in rMDSCs grown on SN for 1, 5, and 9 days. (**a**) SN-60, (**b**) SN-150, (**c**) SN-300, (**d**) SN-500, (**e**) SN-700, and (**f**) glass. The numbers indicate 1, 5, and 9 d. Blue, DAPI; red, actin; green, integrin β1. Magnification, ×400; scale bar = 50 μm. (**d’9**), (**e’9**), and (**f’9**) are enlarged images (magnification, ×1000). (The enlarged actin–integrin β1 immunostaining images are shown in Appendix A).

**Figure 7 nanomaterials-10-01651-f007:**
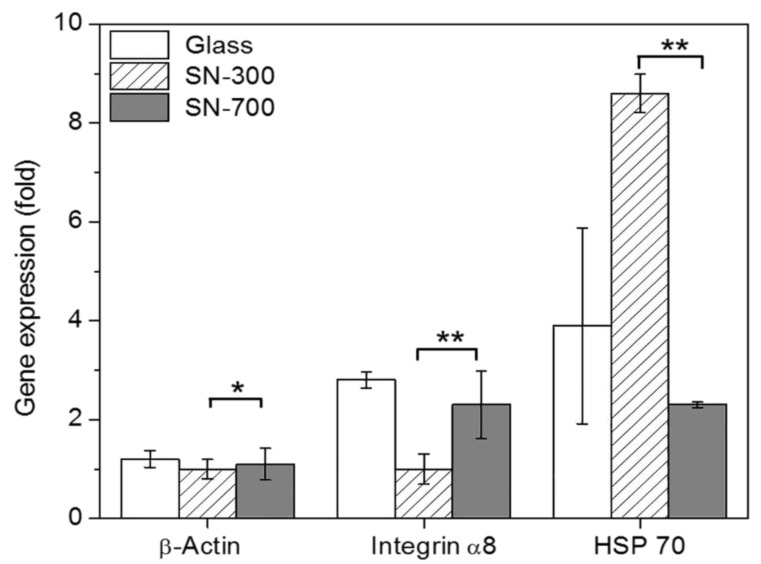
Analysis of β-actin, integrin α8, and HSP 70 mRNA expression by quantitative RT-PCR. The expression values of β-actin, integrin α8, and HSP 70 mRNA in rMDSCs grown on glass, SN-300, or SN-700 at 5 d were normalized to those of GAPDH expression of rMDSCs on glass, SN-300, or SN-700. Gene expression data were analyzed by one-way ANOVA with Bonferroni’s multiple comparison (* *p* < 0.05, ** *p* < 0.01).

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
