# Peer review of "Behavior of Muscle-Derived Stem Cells on Silica Nanostructured Substrates"

_nanomaterials, 2020, doi:10.3390/nano10091651_

Round 1

Reviewer 1 Report

The present paper deals with the analysis of the role played by nanostructured silica surfaces obtained by Langmuir-Blodgett on the adhesion and proliferation of MSCs. The topic of the paper is of interest provided that its important implications in the biomedical field and, in particular, in tissue engineering and regeneration.

However, in the introduction section it is not completely established the novelty of the manuscript in my opinion, in particular, regarding previous manuscripts of the authors and in relation to other previous papers in the field. Then, it would be desiredable to better stress this point.In addition, other important issues should be elucidated prior consideration to publication:

  1. Lines 35 and 36 in the introduction section are almost a pure repetition of previous lines.
  2. Line 82: Authors stated that they prepared silica NPs of 900 nm in size. However, these are not mentioned at all in the rest of the manuscript.
  3. Section 2.3: the description of how authors seed the MSCs cells is unclear. Please, rewrite lines 101 to 105 to make it much clearer.
  4. Line 115: What is Va induction medium? Which is the role here?
  5. lines 124-125. It is not completely clear how authors measure the MSC proliferation, that is, they transferred a partial volume of the formazan crystals solubilised previously in 24 wells
  6. Section 2.5, lines 135-139: please rewrite the immunostaining protocol, which is unclear with some mistakes.
  7. Section 3.2: How the comparative percentages of cell proliferation were calculated? Because in terms of the optical density these are much lower for cells on nanostructured substrates than in the crystal control

Author Response

Responses to the comments by Reviewer 1

We appreciate the reviewer’s comments. We have addressed each of these comments below and have highlighted revisions made to the relevant passages of the manuscript.

1) Lines 35 and 36 in the introduction section are almost a pure repetition of previous lines.

We revised two paragraph into one paragraph. 

2) Line 82: Authors stated that they prepared silica NPs of 900 nm in size. However, these are not mentioned at all in the rest of the manuscript.

We appreciate the reviewer’s comment. We made mistake. We prepared silica NPs up to 700 nm. So we deleted the 900 nm.

3) Section 2.3: the description of how authors seed the MSCs cells is unclear. Please, rewrite lines 101 to 105 to make it much clearer.

We appreciate the reviewer’s comment. We revised Section 2.3. 

4) Line 115: What is Va induction medium? Which is the role here?

The reviewer’s point is well taken. We made mistake. We revised Va induction medium into culture media. 

5) lines 124-125. It is not completely clear how authors measure the MSC proliferation, that is, they transferred a partial volume of the formazan crystals solubilised previously in 24 wells

We appreciate the reviewer’s comment. We revised the unclear sentence.  

6) Section 2.5, lines 135-139: please rewrite the immunostaining protocol, which is unclear with some mistakes.

The reviewer’s point is well taken. We revised unclear part of Section 2.5.

7) Section 3.2: How the comparative percentages of cell proliferation were calculated? Because in terms of the optical density these are much lower for cells on nanostructured substrates than in the crystal control.

The reviewer’s point is well taken. We performed the rMDSCs cultured were monitored optical density using MTT assays and calculated as the relative cell numbers by comparison with control (Glass). We revised section 2.4.

We again appreciate the reviewer’s useful suggestions and comments. We have revised the manuscript to be in line with the reviewer’s comments as much as possible. We would again appreciate your kind consideration to our paper’s publication in Nanomaterials.

Reviewer 2 Report

Kim and collaborators in this paper address an interesting topic: how the silica nanostructured substrates' properties influence cells biological properties?

They choose muscle derived stem cells as a biological relevant model to make some observations on substrates obtained from differently sized silica nanoparticles.

Both the topic and the experimental setting are interesting, but the final outcome is poor. A major problem is the carity of the data presented. The images are really small and hard to understand. Probably it would be more clear if they show in the final paer only a selection of the images and show the others as supplemental material.

Fig 4 needs a colorimetric scale to help understand of the cell pictures. IAlso the graph is  confusing: do SN 60 and 150 completely overlap? And if it was not possible to measure cell size at day 1 and 2, why the line is set at zero? Moreoven in the caption the use of "upper" and "lower" is confusing.

Immunogluorescence images (figure 5 and 6) are too small and a clearer legend is neede: what do the different colors represent? Moreover in the material section it is not stated that the secondary antibody is labeled with a fluorochrome.

"Genetic responses" have been evaluated by PCR, but the authors seem to mix up RT-PCR (reverse trancriprion PCR, used to obtain cDNA from RNA) and quantitative or real time PCR (qPCR). Please revise and make it clear which techniques did you used. Morever, if you analyse mRNA it would be better to speak about bene expression rather than genetic responses.

How many times did you repeated the experiments? Sometimes it is not clear if you performed three technical replicates or if you performed the experiment three times.

Finally, the discussione should be together with the results, but actually the results are discussed only in a superficial level.For example, do the authors have any hypothesis to explain the effect of SN-300 on HSP-70?

Author Response

Responses to the comments by Reviewer 2

We appreciate the reviewer’s comments. We have addressed each of these comments below and have highlighted revisions made to the relevant passages of the manuscript.

1) A major problem is the carity of the data presented. The images are really small and hard to understand. Probably it would be more clear if they show in the final paer only a selection of the images and show the others as supplemental material.

We added the enlarged images into the Supporting Information Figures S1-S5. 

2) Fig 4 needs a colorimetric scale to help understand of the cell pictures. I Also the graph is confusing: do SN 60 and 150 completely overlap? And if it was not possible to measure cell size at day 1 and 2, why the line is set at zero? Moreoven in the caption the use of "upper" and "lower" is confusing.

We think that the reviewer’s comment is well founded. We added the colorimetric scale and histogram graph of the height and length of rMDSCs grown on different SN to help understanding of the cell picture at all days in Figure 4. Additionally, we revised the caption of Figure 4.

3) Immunogluorescence images (figure 5 and 6) are too small and a clearer legend is neede: what do the different colors represent? Moreover in the material section it is not stated that the secondary antibody is labeled with a fluorochrome.

We added the enlarged images into the Supporting Information Figures S2-S5. In caption of Figures 5 and 6, we revised and added the immunostaining protein represented by each color. Additionally, we revised in the experimental section 2.5. 

4) "Genetic responses" have been evaluated by PCR, but the authors seem to mix up RT-PCR (reverse trancriprion PCR, used to obtain cDNA from RNA) and quantitative or real time PCR (qPCR). Please revise and make it clear which techniques did you used. Morever, if you analyse mRNA it would be better to speak about bene expression rather than genetic responses.

The reviewer’s point is well taken. We revised in the experimental section 2.6. Furthermore, we revised genetic responses to genetic expression thoroughly in the manuscript.

5) How many times did you repeated the experiments? Sometimes it is not clear if you performed three technical replicates or if you performed the experiment three times.

We performed three times for each experiment. So, we added the explanation in experimental sections. 

6) Finally, the discussione should be together with the results, but actually the results are discussed only in a superficial level. For example, do the authors have any hypothesis to explain the effect of SN-300 on HSP-70?

The reviewer’s point is well taken. We added the discussion associated with results. Meanwhile, as mentioned in the introduction, rMDSCs can induce the stress when attached and proliferated onto a surface. In this work, we confirmed that fewer rMDSCs adhered to the surface with low roughness, resulting in a lot of stress in rMDSCs to attach into the surface of SN-300. We described the explanation in introduction and section 3.3.

We again appreciate the reviewer’s useful suggestions and comments. We have revised the manuscript to be in line with the reviewer’s comments as much as possible. We would again appreciate your kind consideration to our paper’s publication in Nanomaterials.

Reviewer 3 Report

  1. That would be excellent if the author could provide AFM pics for all surfaces (SN-60, SN-150, SN-300, SN-500 and SN-700) in supplementary information.
  2. Based on current cell data, could author further analysis of the morphological parameters (aspect ratio, cell length, and nuclear shape Index) of rMDSCs on the different surface?
  3. It will be an interesting result if author could provide rMDSCs behavior on the surface prepared by combination of SN-150 and SN-500.

Author Response

Responses to the comments of Reviewer 3

We appreciate the reviewer’s comments. We have addressed each of the comments below.

1) That would be excellent if the author could provide AFM pics for all surfaces (SN-60, SN-150, SN-300, SN-500 and SN-700) in supplementary information.

The reviewer’s point is well taken. We added the AFM images for all surfaces in Supporting Information Figure S1. 

2) Based on current cell data, could author further analysis of the morphological parameters (aspect ratio, cell length, and nuclear shape Index) of rMDSCs on the different surface?

We think that the reviewer’s comment is well founded. We added the histogram graph of the height and length of rMDSCs grown on different SN to help understanding of the cell picture at all days in Figure 4, although we did not analyzed other parameters of rMDSCs.

3) It will be an interesting result if author could provide rMDSCs behavior on the surface prepared by combination of SN-150 and SN-500.

The reviewer’s point is well taken. We did not perform the experiment in current work. However, we will perform the suggested experiment using combination of SN-150 and SN-500 as future studies. We described the future additional work in the conclusion part.

We again appreciate the reviewer’s useful suggestions and comments. We have revised the manuscript to be in line with the reviewer’s comments as much as possible. We would again appreciate your kind consideration to our paper’s publication in Nanomaterials.

Round 2

Reviewer 1 Report

After changes made, the paper is now acceptable for publication.

Author Response

Responses to the comments by Reviewer 1

We appreciate the reviewer’s useful suggestions and comments in the previous revision. We would again appreciate your kind consideration to our paper’s publication in Nanomaterials.

Reviewer 2 Report

The authors revised the paper according to reviewers' suggestions. The quality of the paper has been improved, but some monor corrections are still required:

  • "genetic expression". The authors replaced the terms "genetic responses" with "genetic expression". It's better, but the correct expression is "gene expression" sinche genetic expression  is confusing: genetic reminds to DNA, expression to RNA... gene expression is much better. Moreover they wrote that gene expression was evaluated by RT-PCR. Actually, they performed RT-PCR to obtain cDNA from mRNA, but the quantification was made with a quantitave (aka real time) PCR, in fact they used SYBR green and analysed the results with delta Ct comparitive method. So they performed a qPCR to obtain quantitative data. If they want to mention RT-PCR, it could be said that they performed RT-PCR followed by qPCR
  • In the M&M section, when describing MTT assay, there is an incomplete sentence (page 3 line 140). Moreover, the last sentence of the paragraph is not clear.: "The relative cell number was converted from optical density of each SN based on the optical density of
    glass (Control)". Do they mean that data were express as percentage over control, represented by cells cultured on glass? Plese rewrite the sentence
  • Immunofluorescence M&M. Again, plese be more specific on the secondary antibody you used: the images show the immunocomplexes in green, so the secondary antibody should be conjugated with a green fluorochrome (FITC? Alexa488?) Please specify.
  • statistical analysis: please specify the values for statistical significancy (p<0.05 as usual or something different?)
  • figure 4: no marks for p values are shown. Does it means that differences have no statistical significance or statistical anlysis has not been performed?

Author Response

Responses to the comments by Reviewer 2

We appreciate the reviewer’s comments. We have addressed each of these comments below and have highlighted revisions made to the relevant passages of the manuscript.

1) Genetic expression

We revised genetic responses to gene expression thoroughly in the manuscript.

2) MTT assay

We revised the highlighted sentences in MTT part.

3) Immunofluorescence

We revised the highlighted sentences in immunofluorescence part.

4) Statistical analysis

We revised Figure 4 with statistical analysis.

We again appreciate the reviewer’s useful suggestions and comments. We have revised the manuscript to be in line with the reviewer’s comments as much as possible. We would again appreciate your kind consideration to our paper’s publication in Nanomaterials.